# A Comparison of Head Movement Classification Methods

**DOI:** 10.3390/s24041260

**Published:** 2024-02-16

**Authors:** Chloe Callahan-Flintoft, Emily Jensen, Jasim Naeem, Michael W. Nonte, Anna M. Madison, Anthony J. Ries

**Affiliations:** 1U.S. Army Combat Capabilities Development Command (DEVCOM) Army Research Laboratory, Aberdeen, MD 21005, USA; anna.m.madison2.civ@army.mil (A.M.M.); anthony.j.ries2.civ@army.mil (A.J.R.); 2Department of Computer Science, University of Colorado Boulder, Boulder, CO 80303, USA; emily.jensen@colorado.edu; 3DCS Corporation, Alexandria, VA 22310, USA; jnaeem@dcscorp.com (J.N.); mnonte@dcscorp.com (M.W.N.); 4Warfighter Effectiveness Research Center, United States Air Force Academy, Colorado Springs, CO 80840, USA

**Keywords:** virtual reality, head movement, movement classification

## Abstract

To understand human behavior, it is essential to study it in the context of natural movement in immersive, three-dimensional environments. Virtual reality (VR), with head-mounted displays, offers an unprecedented compromise between ecological validity and experimental control. However, such technological advancements mean that new data streams will become more widely available, and therefore, a need arises to standardize methodologies by which these streams are analyzed. One such data stream is that of head position and rotation tracking, now made easily available from head-mounted systems. The current study presents five candidate algorithms of varying complexity for classifying head movements. Each algorithm is compared against human rater classifications and graded based on the overall agreement as well as biases in metrics such as movement onset/offset time and movement amplitude. Finally, we conclude this article by offering recommendations for the best practices and considerations for VR researchers looking to incorporate head movement analysis in their future studies.

## 1. Introduction

A central goal of human research is to understand behavior as it exists in the real world. With the advent of head-mounted displays (HMDs) in virtual reality (VR) and other sensor systems, this goal is more attainable than ever. These systems allow for increased flexibility to the participant in the allowance of eye, head, and torso movement [1], while maintaining the experimenter’s control over stimulus presentation and data collection. Specifically, researchers now have the unprecedented ability to quantify and characterize head movements, a key component of how humans explore and navigate the natural environment. However, with the introduction of any new capability, there is also the need to develop tested methods for characterizing and analyzing the produced data. The present study explores the head movement field, presenting multiple methods of classifying head movements and comparing them to one another for the purposes of organizing a collection of the best practices for future researchers collecting this data type.

This effort is akin to that in the field of eye tracking, where classification algorithms have been formalized, compared, and made available to the community at large [2]. In part because of this effort, eye tracking data have been used to understand how the visual system processes complex information [3,4] as well as provide insight into co-occurring executive functions [5,6]. The bulk of this study has been conducted using restricted, desktop displays. However, it is imperative to remember that, in real-world vision, eye movements often occur in the context of head movements [7]. This is important not only because eye movements sometimes occur in compensation with head movements (e.g., the vestibulo-ocular reflex) [8], but also because head movements in their own right provide insight into cognitive processes [9,10] as well as individual differences in behavior [11]. For instance, head movements support exploratory search [10] and are affected by the available visual short-term memory resources [12]. Moreover, head movements show unique changes from eye movements in tasks such as reading when familiarity of the text is manipulated [13]. Concurrently, these results demonstrate that characterizing and analyzing head movements offer an important complement to not only the eye tracking literature but also more generally contribute to the field of visual cognition.

From an applied standpoint, given the aforementioned results, head movements have the potential to inform intelligent or adaptive technologies of a human operator’s current state or task goals. A key benefit of this modality is that the information is easily attainable with sensors that may be more robust and cause less interference with the operators than other proposed human-sensing modalities such as an electroencephalogram (EEG) or even eye tracking. As such, analyzing head movements has the potential to not only inform our understanding of how vision occurs in the natural environment but also provide information for spatial computing applications and future augmented reality display technologies. However, to maximize the utility of this information, it is essential that the methodologies to analyze such data are properly established.

Prior research has incorporated and investigated head movements in conjunction with cognitive function [9,13]. However, there is yet to be an standardized methodology for what constitutes a head movement or how to identify an epoch of data pertaining to head movement. This is an essential step in being able to compare metrics such as the velocity, duration, direction, and amplitude of head movements across experimental conditions or between studies. To that end, the current study uses a data set collected for the purposes of characterizing variability in eye and head movements in a virtual environment using an HMD. In this task, participants oriented toward targets, which moved smoothly or instantaneously to peripheral locations, to elicit head and eye movements of varying velocity profiles across a range of eccentricities and directions. As such, this paradigm provides a unique data set by which we can test multiple approaches for head movement classification, and like algorithm comparisons in the eye movement literature, compare their performance to human raters (e.g., [14]). The goal is to present researchers with a set of best practices when analyzing head movement data as well as guidelines on the potential auxiliary data streams that may help with movement classification.

## 2. Materials and Methods

### 2.1. Ethics Statement

This experiment was approved by the Institutional Review Board at the U.S. Air Force Academy (USAFA) and U.S. Army Combat Capabilities Development Command (DEVCOM) under Project Number ARL 21-132. All procedures were in accordance with the Declaration of Helsinki.

### 2.2. Participants

Twenty people participated in this data collection. To provide a ground truth data set, two human raters had to manually code when head movements occurred. As this was a labor-intensive process, data from six participants were randomly selected (one female, average age of 33 years) to be analyzed in the current study. This down selection of sample size for manual coding is common in the eye movement labeling literature to build ground truth data sets (e.g., [15,16]). Participants were United States Air Force Academy cadets recruited through Sona Systems subject pool. They received course credit for participation, and volunteers were recruited through flyers and email announcements, receiving no remuneration. Prior to experimentation, all participants provided written informed consent and completed Snellen chart (at least 20/40) and Ishihara color plates to confirm they had normal vision or corrected vision.

### 2.3. Apparatus

The experimental paradigm was designed using the Unity gaming engine (Unity Technologies, Figure 1A). The HTC Vive Pro Eye VR headset (1440 × 1600 pixels per eye, 90-Hz refresh, 110° field of view, VIVE SRanipal SDK) with integrated eye tracking from Tobii Technologies (120-Hz sampling rate, Tobii XR SDK) was used to present stimuli using a Corsair One PC (Windows 10, Intel Core i9 CPU @ 3.6GHz, 64-bit, Nvidia GeForce RTX 2080Ti, 32 GB RAM), and two external lighthouses were used for tracking head and torso position. Participants were provided instructions and practiced correctly positioning the VR headset prior to experimentation while seated in a fixed position chair.

Torso tracking, using a Vive tracker on the chest, as well as eye tracking and EEG data were collected but are outside the purview of this paper and so are not discussed further. The synchronization of all data streams was accomplished using Lab Streaming Layer (available here: https://github.com/labstreaminglayer/LSL4Unity, accessed on 15 February 2024), a network-based recording software designed to integrate multiple data streams with sub-millisecond precision [17].

### 2.4. Calibration

The Tobii SDK calibration system was used at the start of every experimental block. A three-dimensional virtual volume was fixed to the participant’s head to isolate eye movements from head movements during calibration. Participants were told to fixate sequentially on the five-point calibration (center and upper/lower left/right regions) screen with calibration points placed 1.2 Unity meters from the participants (creating a calibration point approximately 0.5° of visual angle in diameter). After a successful eye tracking calibration (based on Tobii software feedback), the participants’ current position (both eye and head position) was used for the subsequent initiations of trials.

### 2.5. Procedure

Participants first read task instructions and, prior to starting the experiment, achieved at least 80% accuracy on 10 practice trials. On each trial, participants began by centering their gaze on a light gray disk (5° dva in diameter) with a small back ‘O’ symbol in the center for 500 ms to begin the trial. Blinks or significant gaze deviations would result in a reset of the 500 ms clock. After this stationary gaze period, the ‘O’ symbol disappeared, and the disk moved to a random eccentricity (e.g., 10, 20, 30, 40, or 50° dva from the center) in a random direction (e.g., 0, 45, 90, 135, 180, 225, 270, or 315°). In the Pursuit condition, the disk moved smoothly at 20°/s in a straight trajectory. In the Instantaneous condition, the disk disappeared from the center and appeared at the outer location. These eccentricities were chosen in order to keep all stimuli in the Instantaneous condition within the field of view reported by the VR manufacturer. Pursuit and Instantaneous conditions were included to elicit a wider range in the types of head movements performed. In both conditions, once the disk reached this eccentricity, a Gabor patch with either low (0.5 cycles/degree) or high (4.9 cycles/degree) spatial frequency at 100% contrast, tilted 3° either to the left or right, appeared on the disk for 1000 ms (Figure 1B). Participants were told to respond as quickly and as accurately as possible, irrespective of the Gabor patch being tilted to the left or right using the left and right controller buttons, respectively. Pursuit and Instantaneous conditions were blocked and counterbalanced across participants. Participants complete three blocks of each condition (128 trials per block, 640 trials in total). Gabor spatial frequency and tilt, as well as eccentricity and the direction of motion, were intermixed within each block.

### 2.6. Head Movement Classification Methods

The HMD used here returns the x, y, and z position coordinates of the head as well as the rotation of the head around each axis at every time sample (Figure 1C). For head movement classification, the methods mentioned below used some combination of the head’s angular speed, calculated by the angular difference over time (degrees/second), the head acceleration (the derivative of head speed, degrees/second^2^), and the head angle magnitude, the difference in angle between the head’s current position and a forward-facing vector (degrees).

#### 2.6.1. Human Raters

Five human raters were used to manually code head movement data to test algorithm accuracy. Data files were randomly assigned to the rater such that every subject’s data were coded by two different raters. Human raters had an average of three years’ experience collecting and analyzing eye/head tracking data. To perform this, a graphical user interface (GUI) was developed (see Appendix A) that displayed both head angular speed as well as head angular magnitude across time. Human raters were instructed to click on time points when they believed head movements began and ended. While raters identified head movements throughout the entire block of experimental trials, to make the scope of study more manageable, they were only tasked with classifying head movements between the start of a trial (the onset of a center fixation point) and the end of the trial (the time at which a participant made a response to target) for two blocks of trials (one block of Instant trials, one block of Pursuit trials). Within the GUI display, the time at which the trial began, the time at which the Gabor patch appeared, and the time at which the participant responded to the target Gabor were indicated by green, yellow, and red diamonds, respectively (Figure 2).

#### 2.6.2. Baseline Movement Adaptive Threshold (BMAT)

This algorithm was inspired by Engbert and Kliegl’s [18] saccade classification algorithm for eye movement data. For each participant, a three-dimensional ellipse is constructed by setting a threshold of *n* standard deviations from the mean angular head speed in the x, y, and z directions, independently. This ellipse then serves as a threshold to classify motion. Specifically, for this algorithm, the data used to construct the threshold are taken from the 500 ms window before the start of the trial. In this way, we build a threshold for when the head is “moving” based on a window of data where the participant is instructed to keep their head stationary and maintain gaze on a centered disk for 500 ms to begin each trial. This model was fitted iteratively, varying *n* (the number of standard deviations from the mean that the threshold is set) over a range of 0.2 to 4.0 to find the best fit.

#### 2.6.3. Smoothed Velocity Threshold (SVT)

This classification algorithm was adapted from a saccade detection algorithm [19]. First, head angular speed is calculated, and noise is removed with a Savitzsky–Golay finite impulse response filter. A speed threshold is then applied to define movement events. To obtain the onset and offset times of a head movement, the algorithm walks forward and backward in time from each speed peak over threshold and marks movement onset and offset times where the head’s angular acceleration was zero.

#### 2.6.4. Chen and Walton

This classification algorithm was adapted from Chen and Walton’s [20] classification of macaque head movements. A sliding 100 ms window is applied to the head’s angular speed data. Motion onset is identified as when at least 72% of data points in that window are above threshold (6°/s), and there are fewer than three consecutive time points, where the head speed is below threshold. The exact time of motion onset is recorded as the first time point within a qualifying window that is above threshold. Once motion onset is established, motion offset is defined similarly using a 22 ms window where 72% of data points are below threshold. The exact time point of offset is defined as the first time point within a qualifying window under threshold.

#### 2.6.5. Differential IMU-like Zero Crossing Observation (DIZCO)

This algorithm first calculated head angular acceleration in both the x and y directions (the left–right and up–down directions, respectively, see Figure 1). This acceleration data were then filtered below 10 Hz. Zero crossings in either axis’ data correspond then to changes in directions for the head rotation in that respective axis. To reduce the effect of noise, a threshold is set to disregard changes below a certain magnitude. This threshold is set initially through a process of trial and error, by comparing the algorithm classifications with event markers in the data set for trial starts and Gabor pattern presentation, to set the threshold for results that were consistent with the segments of time where head movements and resting periods were likely. The ideal threshold ignores minute fluctuations in head angular acceleration data but is sensitive enough to recognize a larger change in instantaneous velocity. Each segment of time in the data stream is then assigned a movement direction based on the combination of calculated slopes for each axis, which classifies each portion of data as a movement of the head in either one of four cardinal directions, one of four orthogonal directions, or in the case of no movement being detected, a baseline resting state. In this way, the DIZCO method differs from the other methods as it accounts for and classifies a direction for each head movement rather than just labeling time windows where movement occurs. The threshold is validated by plotting the fraction of variance explained (FVE) in the EEG stream by head movements over the scalp electrodes, showing low FVE for times labeled as rest periods.

#### 2.6.6. Decision Tree

A supervised machine learning algorithm was used to label each data point based on the raters’ original labels. Although the algorithms presented previously generate onset and offset times for head movement intervals, machine learning approaches generate a label for each time step in the recorded head movement data. After training the model, the generated predictions were processed to create similar onset and offset times to compare with previous methods. We discarded any intervals that were less than half the length of the average interval length coded by the raters. The model was implemented in Python, using the Scikit-learn [21] library and was a decision tree, which aimed to learn simple decision rules to classify the data [22]. For each label in the training set, the input features were the smoothed head speed and head angle at the current time step. The model used the Gini impurity criterion to evaluate potential decision rules and was limited to a maximum depth of 10. We evaluated the model using five-fold cross validation, which achieved an average accuracy of 57.5%. While more sophisticated methods (i.e., neural networks [23]) may be used to identify head movements, we chose to use the decision tree as it provides interpretable results and is similar to the velocity threshold algorithms discussed previously.

### 2.7. Metrics of Algorithm Comparison

The current paradigm was designed specifically to elicit a wide range of head movements by moving the tracked stimulus (the disk) in two different ways (smoothly and instantaneously) as well as in a variety of directions for varying eccentricities. Instant trials are likely to elicit faster, more ballistic-like head movements compared to Pursuit trials, which are more likely to produce sustained movements. Algorithms may exhibit varying performance across different motion profiles, as demonstrated in the case of saccades and smooth pursuits (e.g., [24]). Furthermore, they may display differential sensitivity to minor and major head movements in distinct directions. To evaluate the agreement between algorithms and human raters, assessments were conducted considering different motion types, directions, and eccentricities.

#### 2.7.1. Cohen’s Kappa

Borrowing from the eye tracking literature, Cohen’s kappa (*K*) was used to score a sample-by-sample agreement between the various algorithms tested and the human raters [2,25]. This metric compares the relative observed agreement (*Po*) with the agreement expected to be observed by chance (*Pe*) between two raters [14,15].
*K* = (*Po* − *Pe*)/(1 − *Pe*)

*K* then can range between 1 (perfect agreement) and 0 (chance agreement). Each of the three thresholding models was run on each participant’s data with a range of threshold values. For every threshold value tested, the kappa values generated by comparing the model output to the human ratings were averaged within participant (to obtain one kappa value from the two human raters of each participant). Finally, these kappa values were averaged across a subset of five participants. The threshold that yielded the highest mean kappa was then used to classify the head movements of the remaining participant. This leave-one-out process was repeated for each participant, and the mean kappa and standard error were reported. Additionally, the average of the best fitting threshold for each subsample as well as the standard error were also reported.

#### 2.7.2. Other Agreement Metrics

The bias in onset time, offset time, duration, and amplitude of head movement were also reported. This method was adapted from a previous algorithm classification study [26] to provide a fuller picture in how these algorithms classify the windows of movement. To achieve this, a ground truth data set was created by combining the classifications of both human raters. For instance, if a given time point in the series had been classified as the head moving by either human rater, that time point was classified as movement. Consecutive time points classified as movement were then considered a single head movement. This aggregation was considered as the ground truth. The next step was then to match a certain algorithm’s classification to a movement in the ground truth. If the midpoint of a ground truth head movement was contained within a movement time window of the given algorithm, those two movements were judged as the same movement. Only one-to-one pairs of movements were kept for analysis. In other words, if multiple ground truth movements’ midpoints were contained in a single algorithm classified movement, that movement was removed from analysis as it had no one-to-one pairing with the ground truth data.

## 3. Results

For Instant trials, human-to-human agreement had an average kappa of 0.62 (*SE* = 0.05) compared to a mean kappa of 0.74 (*SE* = 0.04) in Pursuit trials. Generally, kappa ranges of 0.01–0.20 indicate no agreement to slight agreement, 0.21–0.40 is fair agreement, 0.41–0.60 is moderate agreement, 0.61–0.80 is substantial agreement, and a kappa value above 0.81 indicates near perfect agreement [27]. As the BMAT, SVT, and Chen and Walton algorithms were all variations of thresholding algorithms, a wide range of thresholds were tested for the optimal performance. For every possible subset of five out of the six subjects classified, the threshold that produced the best average agreement between human and algorithm classifier was selected and used to classify head movements in the sixth subject’s data. This kappa, generated by the sixth subject’s data and the threshold selected were then stored. In this way, a leave-one-out method was used to calculate an average and standard error for both the kappa and thresholds used. As the decision tree and DIZCO algorithms have more parameters, the fitting methods are described in their respective sections of the methods. These models were then fitted to each individual subject, and an average performance was calculated. Kappa values and threshold values (for the thresholding algorithms) are listed in Table 1. An example trial along with the head movements classified by each algorithm is shown in Figure 3.

The correlation between the peak velocity and the amplitude of head movements was calculated, classically referred to as the main sequence in the eye movement literature [28]. As in the eye movement literature, there was a significant correlation in Pursuit trials as rated by human rater (*r* = 0.67, *p* < 0.001), BMAT (*r* = 0.85, *p* < 0.001), SVT (*r* = 0.48, *p* < 0.001), Chen and Walton (*r* = 0.83, *p* < 0.001), DIZCO (*r* = 0.72, *p* < 0.001), and the decision tree algorithm (*r* = 0.83, *p* < 0.001) (Figure 4). This correlation between amplitude and peak velocity was also seen in Instant trials as rated by human rater (*r* = 0.76, *p* < 0.001), BMAT (*r* = 0.91, *p* < 0.001), SVT (*r* = 0.63, *p* < 0.001), Chen and Walton (*r* = 0.90, *p* < 0.001), DIZCO (*r* = 0.82, *p* < 0.001), and the decision tree algorithm (*r* = 0.89, *p* < 0.001).

Onset, offset, duration, and amplitude bias were calculated between each algorithm and ground truth data for Instant and Pursuit trials (Table 2). Bias was calculated as the human-rated value minus the algorithm-rated value. For example, human raters, on average, placed head movements 2.89 ms after the BMAT algorithm judged them as beginning and rated them ending 21.01 ms earlier.

## 4. Discussion

To understand human behavior, whether for foundational research or a specific application space, it is essential to study that behavior as it arises in the natural environment. For the purposes of experimental control and to isolate specific mechanisms, researchers often use the abstractions of the real world in laboratory experiments. However, the ultimate goal is to relate these findings to real-world scenarios where stimulus input and behavior likely differ. In the field of visual cognition, abstraction often involves restricting the eyes or head to study how the brain processes visual information in isolation of these movement systems. While these experiments have yielded key insights into cognition, to truly understand how visual information is processed in the real world, experimentation must move toward less constrained, more immersive paradigms. Advances in VR technology not only make such immersive paradigms increasingly possible, but also facilitate the collection of additional data streams such as head position and rotation.

The ability to track head position has existed for years. However, now that such capabilities are being integrated into VR HMD technologies, synced with eye tracking, the bar of entry for incorporating head movement into one’s experimental pipeline has never been lower. This is an important addition to the already vast amount of literature on eye movements (e.g., [29,30]) as much of real-world vision occurs as a complex combination of eye and head movements. Moreover, head movements have already been linked to a number of cognitive processes ([10,13]). It is therefore imperative to test and develop tools for the classification of these head movements in order to properly analyze this data stream as well as compare results across paradigms.

In the current work, we ran a control study to elicit the various types of head movements. Participants were required to move their head quickly or in a smooth pursuit to targets at a variety of eccentricities and directions from a forward-pointed position. We then compared five different head movement classification algorithms to human raters. The SVT algorithm showed the overall highest agreement to human raters in terms of kappa values. This is a benefit to researchers as it does not require them to incorporate a baseline stationary portion of their task (as with the BMAT) nor does it require an extensive training set of human coded data (as with the Decision Tree). However, the SVT algorithm did have larger onset and offset biases compared to some of the others, potentially due to the temporal smearing of the Savitzsky–Golay filter. Plotting the start and end rotation of the head for each coded movement, the SVT algorithm also shows similarity to human-rated movements with the bulk of the movements beginning at origin with a few corrective moments, seen in the majority of starting positions, being classified in the center or forward-facing position (Figure 5).

With any threshold classification method, its success will be influenced by where the threshold is set. The current data set showed that, while the standard error of the thresholds that produced the best fit was small within conditions, the thresholds were quite different between the Instant and Pursuit conditions. In a more complex paradigm (e.g., where participants are searching for a dynamic environment), it is likely that faster and slower head movements would be intermixed. As such, it is possible that the performance of these algorithms would be different compared to how they are performed here with a more tightly controlled paradigm. This should be taken into consideration when setting the threshold. It may be beneficial to manually review a portion of data to find a threshold with sufficient accuracy, but it is important to be thoughtful about how that portion is selected. This issue is analogous to the challenge in setting appropriate thresholds for eye tracking data to classify smooth pursuit accurately [31]. While a priori knowledge of the velocities of objects in the environment can aid in this when classifying data from tightly controlled (e.g., VR) experiments, this becomes more difficult when data collection moves into more real-world paradigms. Similarly, the head motions elicited in this task were tightly controlled in terms of eccentricity and direction. This was a deliberate choice as it provided a ground truth data set for comparing algorithm classifications. However, in natural vision and navigation, head movements will have such constraints, and researchers will need to be thoughtful in their definition of what constitutes a head movement. For instance, a head movement could be defined as a period of time where the head’s angular speed is above some threshold, as would be classified by the BMAT, SVT, and Chen and Walton. Alternatively, a head movement could be defined by angular speed and direction (e.g., is a person shaking their head ‘no’, one movement or two, repeated?), as would be classified by the DIZCO method.

Another consideration when collecting and analyzing head movement data is that only head rotation was used in the classification algorithms explored here. However, there are potentially other sources that may help refine head movement analysis. Collecting muscle activity in the neck could potentially increase an algorithm’s sensitivity to movement, especially slower velocity movement that may be missed through thresholding. Adding additional features may also improve machine learning approach such as the decision tree shown here.

Finally, depending on the application, it may make more sense to leave head movement data as a continuous variable rather than discretizing it into time windows of movement. Head movements do not have the same ballistic trajectory as saccades. While the head can move at a high velocity from one point to another, the head is also likely to make small, slow compensatory movements as the eye searches the environment. As such, depending on the experimental design, it may make more sense for researchers to compare metrics such as the average rotational velocity or the rate of direction change in a given trial. These metrics would not require the classification of discrete head movement windows but would still provide a characterization of how head movements were employed in a given task.

## Figures and Tables

**Figure 1 sensors-24-01260-f001:**
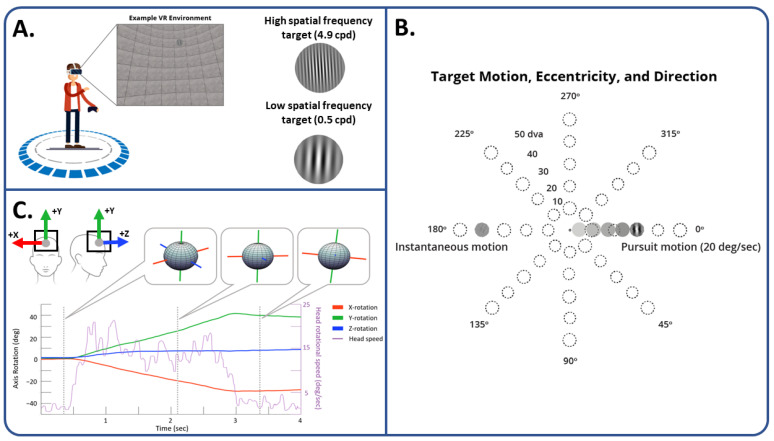
Paradigm schematic and example of data recorded via headset. (**A**) An example of the virtual environment experienced by participants during the search task as well as examples of high and low spatial frequency targets used. (**B**) Potential motion trajectories of target disks. Disks could move smoothly in the Pursuit condition or instantaneously appear at a new location in the Instantaneous condition. The dotted line circles represent potential locations that the disk could either appear at or move to and were not visible in the actual experiment. In the bottom of the panel is an example of the high and low spatial frequency Gabor patches used. (**C**) A graphic depicting how data are collected and analyzed. Unity tracks head position and rotation from the center of the headset using a left-handed coordinate system where the Y-axis is up/down, the X-axis is left/right, and the Z-axis is backwards/forwards. Head rotation data are plotted for each axis during an example trial. This trial was a Pursuit trial where the target smoothly moved in the 225° direction, to an eccentricity of 50° (see panel B). Snapshots illustrate how these data translate into head rotation at the three time points throughout the trial. Using the rotation data around each axis, the angular distance over time can be calculated, providing us with an angular head speed (plotted in purple on the right y-axis of the line graph).

**Figure 2 sensors-24-01260-f002:**
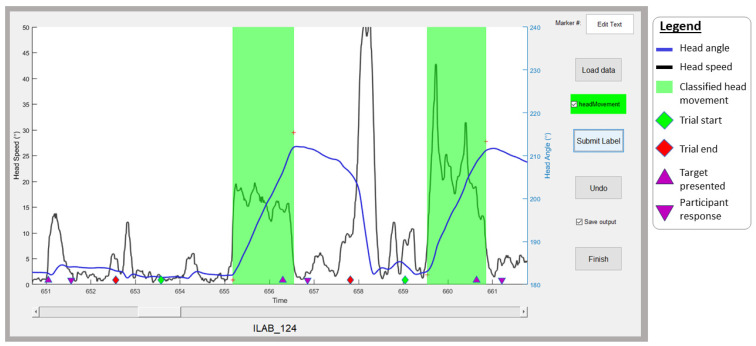
The GUI developed to enable human raters to code head movements. Raters were presented with head angular speed and magnitude over time. Trial information was conveyed through the colored diamond shapes displayed at the bottom of the graph (green diamonds for the onset of fixation, purple triangles for target presentation and response, and red diamonds for the end of the trial). Raters clicked the times at which they judge a head movement as having started and finished. The GUI highlights these windows with green rectangles. The red crosses indicate the positions where the rater clicked to indicate head movement onsets and offsets.

**Figure 3 sensors-24-01260-f003:**
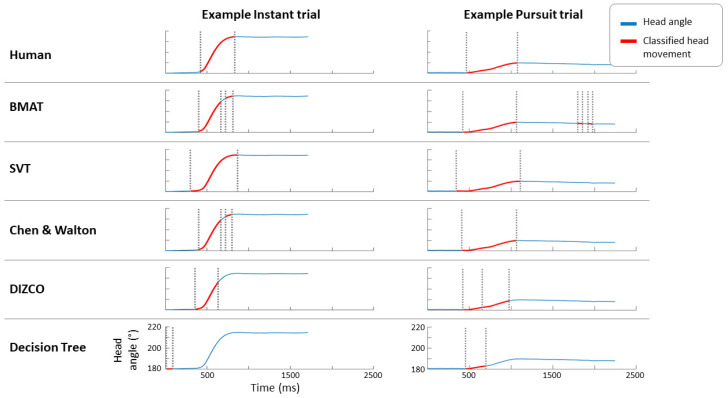
Examples of head angle magnitude over time in Instant (**left**) and Pursuit (**right**) trials. The head angle plotted is the angular difference between the head’s forward vector and world’s forward vector. Time windows where head movements were classified by each method are highlighted in red with onset and offset marked by vertical dotted lines for comparison.

**Figure 4 sensors-24-01260-f004:**
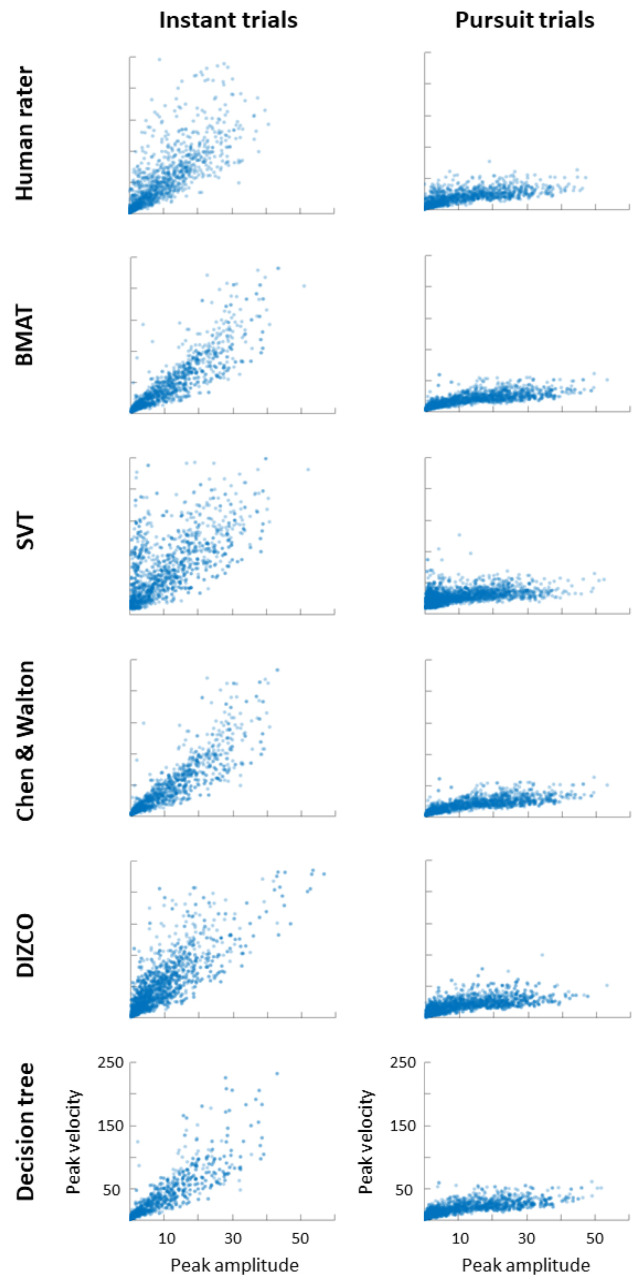
Peak amplitude by peak velocity (also referred to as main sequence plots) of head movements classified by various algorithms for Instant and Pursuit trials.

**Figure 5 sensors-24-01260-f005:**
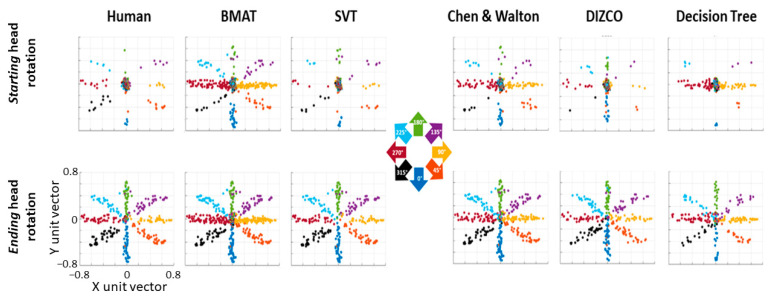
Starting (**top row**) and ending (**bottom row**) head unit vector rotations for every head movement in a trial. Head movements are color coded based on the direction in which the disk moved in that trial.

**Table 1 sensors-24-01260-t001:** Average kappa values for each algorithm. Standard errors are in parentheses. Note that threshold values are only included for the thresholding algorithms (BMAT, SVT, and Chen and Walton). The highest agreement in each trial type has been bolded. As the DIZCO and Decision Tree model have more parameters to fit, a single threshold parameter is not reported.

		BMAT	SVT	Chen and Walton	DIZCO	Decision Tree
Instant Trials	Kappa	0.63 (8.2 × 10^−3^)	0.64 (6.8 × 10^−3^)	**0.65** (6.8 × 10^−3^)	0.34 (6.2 × 10^−3^)	0.48 (0.11)
	Threshold	1.64 (0.03)	18.41 (1.56)	5.62 (0.07)	N/A	N/A
Pursuit Trials	Kappa	0.61 (3.6 × 10^−3^)	**0.71** (4.0 × 10^−3^)	0.62 (3.3 × 10^−3^)	0.39 (0.02)	0.59 (0.06)
	Threshold	1.05 (6.6 × 10^−3^)	9.57 (0.21)	3.49 (0.07)	N/A	N/A

**Table 2 sensors-24-01260-t002:** Average onset, offset, duration, and amplitude biases for each algorithm. Standard errors are included for each metric in parentheses. The algorithm with the smallest bias for each metric, for each trial type, is bolded.

		Onset Bias	Offset Bias	Duration Bias	Amplitude Bias
Instant Trials	BMAT	−2.89 (9.97)	21.01 (18.27)	23.90 (23.91)	−2.41 (1.88)
	SVT	11.04 (23.26)	22.95 (39.92)	11.90 (62.76)	7.65 (5.93)
	Chen and Walton	**−2.72** **(9.46)**	**16.78** **(14.26)**	19.50 (19.07)	−3.16 (1.62)
	DIZCO	20.40 (8.48)	23.43 (15.02)	**3.03** **(20.12)**	8.57 (3.53)
	Decision Tree	−11.80 (16.53)	27.07 (50.10)	38.88 (65.58)	**0.13** **(3.87)**
Pursuit Trials	BMAT	**−1.75** **(11.25)**	27.41 (17.43)	29.17 (22.66)	−2.53 (1.29)
	SVT	76.49 (12.10)	**−2.62** **(20.53)**	−79.11 (23.18)	**−0.77** **(1.87)**
	Chen and Walton	7.73 (13.23)	17.49 (17.30)	**9.76** **(23.43)**	−3.47 (1.22)
	DIZCO	17.02 (9.61)	47.89 (13.27)	30.87 (18.17)	7.87 (2.09)
	Decision Tree	−39.78 (13.62)	61.92 (44.06)	101.70 (54.81)	4.85 (3.01)

## Data Availability

Data are available upon request to the corresponding author.

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
