# Peer review of "A Comparison of Head Movement Classification Methods"

_sensors, 2024, doi:10.3390/s24041260_

Round 1

Reviewer 1 Report

Comments and Suggestions for Authors

This paper presents a comparison of classifying the onset and offset of head movements using IMU data from head-mounted displays. Five established and well-used algorithms were compared against a manual baseline using human coders. The results show clear performance differences between the algorithms. However, there is still room for improvement to reach the performance of human coders.

I would like to thank the authors for their submission. The paper provides a clear and practical premise and investigates it in a well-structed and easy-to-follow way and with clear results. The provided supplemental material containing algorithm implementations will also be beneficial to researchers with less coding experience. I think this paper will be very beneficial to future researchers and practitioners interested in head movement, which is essential in head-mounted AR and VR. A job well done. I only have minor comments that relate to presentational issues and some minor clarifications.

-       Demographics of the 20 participants, and subset of 6 is missing.

-       The VIVE SRAnipal SDK is for eye tracking, not general VR usage. I assume authors are referring to the SteamVR SDK?

-       How many repetitions of each condition did each participant perform? This is currently only indirectly stated from the total number of trials, but could be clearer to save readers having to calculate it themselves.

-       Some figures (Fig 1, 2) and tables (Table 2 are split between pages and should be fixed to increase readability.

-       A legend incorporated in Figure 2 and Figure 3 would increase readability.

-       Fix consistency when referring to axis e.g. x in 2.6.2 but X in 2.6.5

-       What is X and Y in 2.6.5? Please clarify the direction of each axis to avoid any confusion. I assume Unity directions but not all applications follow this convention.

Thanks again to the authors I look forward to seeing this work published. I hope my review was helpful.

Author Response

Thank you so much for taking the time to review our paper. We have made the suggested changes. These comments really help improve the readability of the paper and we appreciate the time and effort you put into this.

Reviewer 2 Report

Comments and Suggestions for Authors

The study, titled "A Comparison of Head Movement Classification Methods," explores various algorithms for classifying head movements in virtual reality (VR) environments. It aims to standardize methodologies for analyzing head position and rotation data, which is increasingly available due to technological advancements in VR. However, this study used a relatively small sample size, which might limit the generalizability of the findings. I have some questions after reading this manuscript. Please comment and revise some issues.

1)      The study focused on specific types of head movements elicited through a controlled VR environment. This may not fully encapsulate the variety and complexity of natural head movements in different real-world scenarios.

2)      The performance of algorithms based on threshold settings could vary significantly based on how these thresholds are determined and applied, which might affect the applicability of the findings in different contexts.

3)      In Lines 92 to 93, information is missing.

4)      Each figure should have its own title. For example, Figure 1 doesn’t have a title, and the figure legend introduces the subfigure right away.

5)      In Lines 180 and 200, the authors need to revise this sentence for the citation format.

6)      The head movements that asked subjects to do must be presented and described clearly in Section 2.2.

Author Response

Thank you so much for reviewing our paper. Below we have addressed each of you comments in turn.

1)      The study focused on specific types of head movements elicited through a controlled VR environment. This may not fully encapsulate the variety and complexity of natural head movements in different real-world scenarios.

2)      The performance of algorithms based on threshold settings could vary significantly based on how these thresholds are determined and applied, which might affect the applicability of the findings in different contexts.

We thought to address comments 1 & 2 together as they both speak to the importance of including nuance in our discussion of head movements and our algorithms performance. This is a really important point that we agree with wholeheartedly and appreciate you calling out as we want to ensure both of these points are stressed to the reader. To address this, we have rewritten a portion of the discussion (lines 409-432) to emphasize both that while the paradigm was specifically designed to elicit head movements at a variety of sizes, directions and speeds, there is the inevitable drawback that these head movements are, by nature of the paradigm, unnatural. We believe that this method still has value in the development of classification algorithms but there are indeed caveats that future researchers should be aware of.

3)      In Lines 92 to 93, information is missing.

This has been added.

4)      Each figure should have its own title. For example, Figure 1 doesn’t have a title, and the figure legend introduces the subfigure right away.

We weren’t entirely sure what you meant by “title”. We rewrote the caption of figure 1 to begin with a summary sentence but weren’t sure if the suggestion was that each figure have a title in the figure itself.

5)      In Lines 180 and 200, the authors need to revise this sentence for the citation format.

 This has been revised.

6)      The head movements that asked subjects to do must be presented and described clearly in Section 2.2.

We were unsure what was meant here or if potentially there was a typo as section 2.2 is the section describing the participants used. We have rewritten the 2.5 to better describe the head movements elicited.